# Petrifyin': Canonical Counter-Discourse in Two Caribbean Women's Medusa Poems

Phillip Zapkin 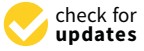

Department of English, Pennsylvania State University, State College, PA 16823, USA; phillipzapkin@gmail.com

**Abstract:** This essay utilizes Helen Tiffin's idea of canonical counter-discourse to read the Medusa poems of Shara McCallum and Dorothea Smartt, two female Caribbean poets. Essentially, canonical counter-discourse involves authors rewriting works or giving voice to peripheral/silenced characters from the literary canon to challenge inequalities upheld by power structures such as imperialism and patriarchy. McCallum's and Smartt's poems represent Medusa to reflect their own concerns as women of color from Jamaica and Barbados, respectively. McCallum's "Madwoman as Rasta Medusa" aligns the titular character from her book *Madwoman* with Medusa to express Madwoman's righteous anger at the "wanton" and "gravalicious" ways of a Babylon addressed in second person. Smartt's series of Medusa poems from *Connecting Medium* explore the pain of hair and skin treatments Black women endure to try and meet Euro-centric beauty standards, as well as the struggles of immigrants, particularly people of color. Both poets claim Medusa as kindred, empowering Medusa as a figure with agency—which she is denied in the Greco-Roman sources—and simultaneously legitimizing both Caribbean literature and the poets' feminist and post-colonial protests by linking them to the cultural capital of the classics.

**Keywords:** Shara McCallum; Dorothea Smartt; Medusa; canonical counter-discourse; classical reception; feminism; post-colonialism; Caribbean poetry; adaptation

## 1. Introduction

What does it mean for African/diasporic writers to reclaim Medusa? What does it mean for feminist writers to reclaim Medusa?

In addition, perhaps more provocatively, does the verb *reclaim* problematically locate ownership of Medusa in a literary/mythic space controlled by authors who are white and male?

Emily Greenwood writes, "A dominant theme in the existing literature is the apparent incongruity of classics in the context of the Caribbean, where the discipline would seem to be on the wrong side of the racial, imperial, and political oppositions that have divided the region historically" (Greenwood 2005, p. 65). However, classics were taught in West Indian education systems, along with British literature, as part of a systematic imposition of British/European culture meant to overwrite the multicultural blending of various West African, Spanish, French, Dutch, and Indigenous traditions alongside British influence. In part due to schools in many Anglophone Caribbean nations retaining variations of British curricula, the Greco-Roman classics have remained popular amongst Caribbean authors into the post-colonial period. However, this popularity is also due to authors' rewriting of classical material to challenge the hierarchical and imperialist assumptions of the colonial period. As Helen Tiffin argues, "contemporary art, philosophies and literature produced by post-colonial societies are not simply continuations or adaptations of European models. The processes of artistic and literary *de*colonization have involved a radical dis/mantling [*sic*] of European codes and a post-colonial subversion and appropriation of the dominant European discourses" (Tiffin 1987, p. 17, original emphasis). Tiffin theorizes what she calls *canonical counter-discourse*, which is a specific way post-colonial authors write back to the

colonial center by adapting materials from the European/imperial canon and turning them into post-colonial properties.

This essay utilizes the idea of canonical counter-discourse to read the Medusa poems of Shara McCallum and Dorothea Smartt, poets from Jamaica and Barbados, respectively. Specifically, I am reading McCallum's "Madwoman as Rasta Medusa" and a series of Smartt's poems from the book *Connecting Medium*, arguing that these poems utilize the Medusa myth to challenge both imperialist and gendered structures of domination.[1] The speakers in McCallum's and Smartt's poems identify with the Medusa myth and its cultural capital in order to challenge patriarchal and racial/colonial hierarchies. In doing so, the poems utilize canonical counter-discourse to promote a liberatory intersectional message. This essay moves dialectically, beginning with a brief biography of Medusa that draws out some of the implications not only of her story but of the nature/process of myth itself. I then turn to how each author treats Medusa, looking closely at the poems themselves. This is followed by sections on how these poems utilize canonical counter-discourse in the feminist tradition to challenge patriarchal dominance and societal expectations of (Black) women, and then how the poems resist colonial/racial hierarchies that have traditionally repressed and delegitimized Afro-Caribbean people. Finally, I conclude with a section on the stakes for intersectionality in canonical counter-discourse and liberatory art.

## 2. Medusa's Biography

Who is Medusa? This is a relatively difficult question to answer accurately because, while some elements of her history are generally agreed upon in Greco-Roman sources, other important details do vary. Then, once one gets to the medieval period and after, authors begin dramatically changing Medusa's story, its meaning, and her very character. In *Medusa: In the Mirror of Time*, David Leeming (2013) gives a thorough analysis of the reception of Medusa in different time periods, showing how interpretations of her change significantly. Garber and Vickers (2003) *The Medusa Reader* collects a wide range of primary sources about the Gorgon—an invaluable resource for anyone studying the reception of the Medusa myth. Here I give only a brief sketch of Medusa's biography as it is presented by classical authors, particularly the accounts in Hesiod, Apollodorus, and Ovid—though later I address modern feminist interpretations in more detail.

In one of the earliest recorded accounts of the Medusa story, Hesiod's eighth–seventh century BCE *Theogony* tells us that she was the child of the pre-Olympian gods Phorcys and Ceto, and sibling of the *Graiai* (the Old Women, sometimes associated with the Fates) and the other two Gorgons (Hesiod 1999, p. 11). For reasons that are never specified—beyond the fact that it must be true for her to be killed in the myth—Medusa is the only one of these divine siblings who is mortal. Hesiod then recounts that "with her the god of the Sable Locks [Poseidon] lay in a soft meadow" so that when Perseus cut off Medusa's head, Chrysaor and Pegasus sprang from her severed neck (Hesiod 1999, p. 11). In the first–second century CE account found in Apollodorus' *Library of Greek Mythology* we learn more about Medusa's appearance, along with that of her Gorgon sisters. Apollodorus describes all three Gorgons as hideous monsters: "The Gorgons had heads with scaly serpents coiled around them, and large tusks like those of swine, and hands of bronze, and wings of gold which gave them the power of flight; and they turned all who beheld them to stone" (Apollodorus 2008, p. 66). Slightly later he mentions in passing that Medusa had angered Athena by claiming to be as beautiful as the deity, which seems to contradict his earlier depiction of the Gorgons as naturally or inherently monstrous—though this may simply be an attempt to provide a plausible motive for Athena's hatred. Apollodorus also elaborates on Perseus' story, describing how the hero came upon her while the Gorgons slept and looked into a bronze shield, rather than directly at her, in order to avoid being turned to stone. The *Library* then recounts Perseus' later adventures, in which he repeatedly uses Medusa's severed head, with its continuing power to petrify, to defeat his enemies before turning the head over to Athena, who fixed it at the center of her shield in order to use that power herself (Apollodorus 2008, pp. 66–67).

When the Roman poet Ovid picks up the story in *The Metamorphoses*, written in 8 CE, he makes major changes from the Greek versions.[2] Ovid's focus is largely on the adventures of Perseus, but at the end of book III, Perseus is asked why Medusa was the only one of the three Gorgons with snakes for hair (Ovid 2005, p. 84)—a divergence from Apollodorus' description, in which all three of the sisters had snake hair. His answer is that Medusa was regarded as a stunning beauty—though no mention here of a comparison with Minerva[3]— and while she turned down many suitors, Medusa was raped by Neptune in Minerva's temple, and "that the deed might be punished as was due, she [Minerva] changed the Gorgon's locks to ugly snakes. And now to frighten her fear-numbed foes, she still wears upon her breast the snakes which she has made" (Ovid 2005, p. 84). In other words, Medusa is punished for being raped in Minerva's temple, and that punishment is to have her beautiful hair transformed into serpents, though presumably along with this comes the power to turn to stone those who look at her.[4]

Of course, *The Metamorphoses* is driven by narratives of transformation, so the idea that Ovid has Medusa changed from a beauty to a monster aligns with his project in the poem. However, he clearly states that (1) Neptune *rapes* Medusa, and (2) this rape takes place inside Minerva's temple. This contrasts both Hesiod and Apollodorus. Hesiod says that Medusa and Poseidon "lay in a soft meadow among the spring flowers" (Hesiod 1999, p. 11), and in *The Library*, Apollodorus simply says that Medusa "had conceived them [Pegasus and Chrysaor] previously by Poseidon" (Apollodorus 2008, p. 66). Importantly, neither of these accounts *precludes* rape, but neither do they make it a definite condition of Medusa's interaction with the sea god. As we later see in more detail, the question of Medusa's agency—i.e., whether she has any—is extremely important in the reception of her myth. The other big change is that Ovid shifts the location. As we saw, Apollodorus seems to struggle to justify why Athena hates Medusa and helps Perseus kill her, but Ovid provides a neat justification in the desecration of the virginal Minerva's temple by sexual violence. These changes—and the fact that they were/are accepted as part of a single coherent myth—suggests that the ancients did not feel any need to be uniform, which is why we see variations in the myth over time and between different authors. It is part of the inherent nature of myth to grow and change as societies grow and change.[5]

Myth's ability to grow and develop with the needs of those who receive it—including McCallum and Smartt in their poetry—raises questions about Medusa's agency in these classical versions. This is important because many modern writers, especially feminists, draw on Medusa for empowerment and self-expression. If, in Greco-Roman versions, Medusa has no agency, this modern tendency to write from her perspective or connect her with empowerment is an important tactic for writing back to the canonical myth. Additionally, what we find when we look at the Hesiod, Apollodorus, and Ovid is that Medusa almost always exists as an object. Her main role in the myth is to have her head cut off by Perseus, who then takes control of the head before giving it to Athena. Furthermore, Ovid's account makes especially clear how little control Medusa has as she is both raped by Neptune and changed into a monster by Minerva—in neither case does Medusa have a say in her fate. The ambiguity of Hesiod's account (and to a lesser extent Apollodorus') opens up the possibility that she might have chosen to sleep with Poseidon, though the phrase "lay with" may be a euphemism for rape, as Greek and Roman gods regularly raped mortal women. However, even if Medusa chose to sleep with Poseidon, Hesiod presents that point as significant only because it leads to the births of Pegasus and Chrysaor when Medusa dies.

Even when Medusa turns her "victims" to stone, little in any of the recorded versions suggests she wants to do so or has any choice—Athena or her birth as a Gorgon simply made Medusa that way.[6] Apollodorus' phrase "they turned all who beheld them to stone" is ambiguous about whether this is an involuntary consequence of looking at the Gorgons, or whether there is some element of will on the Gorgons' part, though I would argue that the word *all* suggests this petrification occurred automatically, especially if the Gorgons themselves did not actually need to look at the person. In contrast to the Greek and

Roman sources, many modern feminist authors present Medusa's terrifying nature as a form of agency, revenge against her rapist displaced to every living thing she encounters.[7] Feminist treatments of Medusa are discussed more below, but perhaps the most direct poetic depiction of Medusa as choosing to transform into a Gorgon comes in Ann Stanford's 1977 "Medusa":

> It is no great thing to a god. For me it was anger—
> no consent on my part, no wooing, all harsh
> rough as a field hand. I didn't like it.
> My hair coiled in fury; my mind held hate alone.
> I thought of revenge, began to live on it.
> My hair turned to serpents, my eyes saw the world in stone. (Stafford 2003, lines 9–14)

There is so much to say about this poem in its own right, but for my purposes the key point is that Stanford's Medusa generates her own change. From her rage, she transforms into the monstrous Gorgon who is capable of choosing and taking vengeance for her violation. Gone, in this poem, is Minerva transforming Medusa as in Ovid; gone is the Gorgons' inherent monstrosity as in Apollodorus. As we see more below, this approach to re-imagining Medusa as a figure with her own will and power aligns with the canonical counter-discourse of both McCallum and Smartt.

### 3. McCallum's Medusa

Jamaican poet Shara McCallum's "Madwoman as Rasta Medusa" is published in her collection *Madwoman*, which deals with themes of dislocation, powerlessness, gender dynamics, and family life. Right from the beginning of the poem, it is clear that the Madwoman-speaker takes on the role of an empowered and vengeful Medusa, deeply imbedded in Rasta theology. The opening lines of the poem are, "I-woman go turn all a Babylon to stone./I-woman is the Deliverer and the Truth" (McCallum 2017, lines 1–2). These lines express a righteous fury rooted in the Rasta opposition to Babylon—a multi-faceted term referring to a wide variety of oppressive power structures, including British colonialism and the politico-economic power systems that suppress global south/Black nations. The poem ends with promises of a divine vengeance carried out by Madwoman, but it can be read as vengeance on behalf of the historically oppressed, colonized, and enslaved peoples of Jamaica:

> I-woman is the Reckoning and Judgment Day.
> This face, etch with wretchedness,
> these dreads, writhing and hissing
> misery, is not the Terror.
> I-woman is what birth from yu Terror. (McCallum 2017, lines 15–19)

For Rastas, the word *dreads* may refer either to the dreadlocks hairstyle or to individuals who wear the hairstyle, particularly Rastafarians. As we will see with Smartt's poems, the link between African/African diasporic women's hair and the Gorgon's snaky locks is a common trope in Medusa poetry. At the same time, if we read McCallum's reference to dreads as a synonym for Rastas, then the line can refer to (primarily Black) Jamaican resistance to British/European culture and colonial authority.

Even as McCallum positions Madwoman-Medusa as a destroyer with the agency to avenge wrongs—"Look pon I/and know what bring destruction" (McCallum 2017, lines 5–6)—the poem simultaneously accuses the "yu" addressee of creating the supposed monster through greed and exploitation. The most direct expression of this indictment is:

> Yu say I-woman is monstrosity
> but is yu gravalicious ways
> mek I come the way I come.
> Is yu belief everyone exist fi satisfy
> yu wanton wantonness. (McCallum 2017, lines 7–11)



The "yu" is gravalicious and wanton—selfish and lascivious. The "yu" wants to consume, to exploit, to possess. Because "yu" is not clearly identified as a single, specific subject, there are many possible interpretations that fit. "Yu" may refer to an individual man or to the patriarchy as a social system; the speaker, after all, repeatedly identifies as "I-woman," with a clear gendered element to her righteous anger. The poem re-emphasizes this theme of female and/or post-colonial anger in the closing lines, quoted in the previous paragraph. This may be Medusa speaking back to Poseidon, Perseus, and/or Athena—though of those three I would favor Poseidon as the subject because, at least in Ovid's version, he is the most obviously wanton and gravalicious. This may also be Madwoman speaking back to patriarchy (or possibly to a specific individual man who has wronged her), because patriarchy positions all women as objects to be possessed—in multiple senses—by men. Conversely, this may be Madwoman as Jamaican speaking back to the colonial and racial hierarchies of Babylon, which position Black West Indians as racially inferior. In my reading, these interpretations are not mutually exclusive, but lend a richness and complexity to the poem as McCallum deploys the Medusa myth to undercut multiple dominant power structures.

Across her work, McCallum utilizes myth to explore thematic concerns such as family relationships, identity, loss, and racial power dynamics. In her debut poetry collection, *The Water Between Us*, three of McCallum's poems draw from Greek mythology: "Persephone Sets the Record Straight" (McCallum 1999b), "the siren's defense" (McCallum 1999c), and "Calypso" (McCallum 1999d). Additionally, the poem "jack mandoora me no choose none" (McCallum 1999a) combines European fairy tales with African/Afro-Caribbean Anansi stories. Then, in her 2011 book *This Strange Land*, McCallum returns to the Persephone myth in "My Mother as Persephone" (McCallum 2011b). Additionally, she draws from the Narcissus myth in "My Mother as Narcissus" (McCallum 2011d) and from Homer's *The Odyssey* in the poems "My Mother as Penelope" (McCallum 2011a) and "Penelope" (McCallum 2011c). Her two Penelope poems self-consciously refer to mythology, raising questions about the female speaker's existence apart from male-authored/inspired myths. In "Penelope," the speaker describes Odysseus returning: "After years adrift, you return/wanting to know how I exist/apart from you and your myths" (McCallum 2011c, lines 10–12). In "My Mother as Penelope," the speaker rejects her own mythic status—perhaps a status imposed by Odysseus or Homer—though this is not as explicit as in "Penelope." She writes, "Listen, after years of waiting,/I tire of the myth I've become" (McCallum 2011a, lines 9–10). Beyond the classical references, McCallum regularly draws on Bible stories, folk stories, songs, etc., to construct her poems. She told interviewer Todd Davis that she has been strongly influenced by various types of storytelling, and that intertextuality comes from that inspiration. In particular, incorporating mythic themes, storylines, and characters has helped McCallum "see in a personal story something archetypal or that resonates a wider human experience" (Davis and McCallum 2015, p. 445). Resonating with a wider human experience is certainly evident in "Madwoman as Rasta Medusa," as we have already begun to see. It is a poem with layers of meaning, which align it with both feminist and post-colonial canonical counter-discourse.

## 4. Smartt's Medusa

Dorothea Smartt's collection *Connecting Medium* contains a series of eight Medusa poems, some more directly focused on the Gorgon than others.[8] For this argument, I focus primarily on "medusa? medusa black!," "let her monsters write," "let me land . . . ," and "medusa: cuts both ways," which I think offer the richest examples of canonical counter-discourse.[9] Central to Smartt's Medusa poems are several specific themes, including the identification of African/African diasporic women's hair with Medusa's snakes, silencing, and feelings of social rejection. These themes are interconnected with and inspired by Smartt's own experiences as the child of Bajan immigrants to London. Her poetry often deals with the difficulties of feeling at home as a member of the African and Caribbean diasporas. This is a central theme in poems such as "connecting medium II" (Smartt 2001j)

and "gambian sting" (Smartt 2001b), to take just two instances from *Connecting Medium*. The first poem exemplifies the anxiety about not belonging in Britain, with the speaker recounting a childhood "African" ritual of placing pennies in the mouth of a glass West Indian fish statue to try and cure her homesickness—though it is not clear which home the speaker is sick for: Barbados, Africa, or a Britain where she will feel accepted. "gambian sting" complicates questions of longing for home even more as it recounts a trip to Africa in which a protagonist seeking a connection to her ancestral continent is conned by a group of young Gambians. Smartt's Medusa poems align with the thematic concern about alienation.

Smartt uses hair imagery continually throughout her Medusa poems. Sometimes these are merely passing references, but "medusa? medusa black!" makes explicit how central hair is in the identification between the speaker and Medusa.[10] The poem opens with a direct claim about Medusa's heritage: "Medusa was a Blackwoman,/afrikan, dread" (Smartt 2001g, lines 1–2). Smartt asserts a clear visual link between dreads, a natural hairstyle for many Black people, and the snaky hair of the mythic Gorgon. This snake–dreadlock connection is strengthened later in the poem, with the line "Banish the snake-woman" (Smartt 2001g, line 37), which breaks down ostensible gaps between Medusa and Black women. Here the "snake-woman" is not Medusa the independent figure, but a visual stand-in for dreadlocked women. This is, of course, a complex connection because there is a long tradition of associating Medusa with negative attributes—of animalistic monstrosity, sexual temptation, foreignness/otherness, etc. Many of these same negative traits have been stereotypically linked to African and diasporic women in the service of white supremacy and patriarchal oppression/violence. However, as we shall see in more detail, Smartt is reclaiming the positive aspects of Medusa—independence, strength, self-reliance, creativity, etc. The injunction to "Banish the snake-woman," in fact, tells readers to embrace the snake-woman, which we know because this line comes in a passage ironically parodying beauty advice for Black women to meet Euro-centric standards.

The poem directly critiques the imposition of white beauty standards on African/African diasporic women, often involving painful hair and skin treatments to try and conform to Euro-centric standards of attractiveness:

> Scrub it bleach it operate on it powder it
> straighten it fry it dye it perm it
> turn it back on itself
> make it go away make it go away.
> Scrub it, step smiling into baths of acid
> and bleach it red raw
> peel skin of life-sustaining melanin.
> Operate on it
> Blackskin—lying, useless—discard it powder it.
> Head? Fuck it, wild-haired woman,
> straighten it fry it, desperately burn scalps.
> *Banish the snake-woman*
> the wild-woman
> the all-seeing-eye woman.
> Dye it,
> *remembrances of Africa fast-fadin'*
> in the blond highlights,
> turn us back on ourselves. (Smartt 2001g, lines 26–43, emphasis added)

This lengthy excerpt from the poem highlights the painful experience of trying to meet Euro-centric beauty standards. The almost twenty lines in a fifty-seven-line poem—and these are not the only lines about hair and skin treatments—give it significant weight as a thematic concern. The seriousness of this problem is further illustrated by the graphic nature of some of the imagery, especially that of burning, peeling skin, bleaching, and

operations. Smartt's language presents a horrifying picture of the attempts to destroy "remembrances of Africa" by destroying Black women's natural hair and skin. As Lizabeth Goodman argues, "Smartt shows how expectations about black women's appearance are connected to a sense of self-definition in relation to the cultural values of (white) women's appearance" and often found wanting (Goodman 2003, p. 273). The speaker of the poem makes this explicit, writing, "My hair as it comes/is just not good enough" (Smartt 2001g, lines 46–47). This theme of not being able to find a place in (British) society pervades the Medusa poems.

The link between the Gorgon and the anxiety about immigration is perhaps most overt in "let me land . . . " a poem about crossing the sea to a land where the characters are unwelcome. The opening portion of the poem is full of oceanic imagery, through which "patterns *snake*, rippling through the diaspora" (Smartt 2001d, line 2, emphasis added) before readers are enjoined to imagine arriving at a cold, damp shore that sounds stereotypically English (Smartt 2001d, lines 15–20). The poem ends by identifying Medusa as an immigrant to these clammy shores (Smartt 2001d, lines 34–38), arriving despite the apparent hostility of those already there, whose "eyes flame lava patterns on skins" (Smartt 2001d, line 19). Although Medusa and "Herself" (an unidentified character) arrive in the UK, they are denied a voice: "No words yet, just sounds; wind/sea spray, distant thunder announce them" (Smartt 2001d, lines 34–35). These same concerns are expressed in "let her monsters write." The poem opens with Medusa trying to squeeze herself into boxes where she does not fit, physicalizing the metaphor of trying to conform to unfamiliar social patterns (Smartt 2001c, lines 1–6). She struggles to reduce the scope of "her frilly outstretched/body" to the boxes that seek to constrict her (Smartt 2001c, lines 5–6). However, the solution to this unnatural compression is artistic expression. Medusa seeks a voice. The poet persona implores, "Under hair/let her monsters write/from all sides—ceiling walls floor" (Smartt 2001c, lines 8–10) as the only way to free Medusa's "singsong body/lacy in the night" (Smartt 2001c, lines 12–13). Medusa's inability to speak reflects her alienation, her sense of dislocation, and it is through artistic production that she begins creating space for herself. As we shall see more below, this kind of artistic reclamation is at the heart of canonical counter-discourse.

## 5. Medusa Poems as Feminist Canonical Counter-Discourse

There is a prominent tradition of twentieth and twenty-first century feminist poets reclaiming the Medusa figure/story as symbols of empowerment, thereby writing back to patriarchy by repurposing a traditionally patriarchal narrative. Alicia Ostriker referred to this process as "revisionist mythmaking" (Ostriker 1982)—essentially an approach to adaption theory that balances awareness of the canonical authority of mythical texts with the transformative possibilities offered by rewriting. Ostriker claims revisionist mythmaking occurs when "the figure or tale will be appropriated for altered ends, the old vessel filled with new wine, initially satisfying the thirst of the individual poet, but ultimately making cultural change possible" (Ostriker 1982, p. 72). Her 1982 essay "Thieves of Language" examines this process in American feminist poetry, showing how writers repurpose culturally authorized myths to explore liberatory politics.

One influential thinker who helped conceptualize the reclamation of Medusa was Hélène Cixous, whose famous 1975 essay "The Laugh of the Medusa" advocated that female writers abandon phallocentric writing practices to find ways of writing rooted in women's experience. Cixous exhorts, "She [woman] must write herself, because this is the invention of a *new insurgent* writing which, when the moment of her liberation has come, will allow her to carry out the indispensable ruptures and transformations in her history" (Cixous 1998, p. 1457, original emphasis). For Cixous, Medusa became a symbol of women's silenced voices under patriarchy and masculine forms of writing. By transcending men's writing to discover new feminine/feminist forms of writing that liberate women's authentic voices, Cixous argues that women can join the laughing Medusa—a figure of

freedom and joy. Many female poets and writers directly took up the challenge, picking Medusa as a symbol of this liberatory re-imagination.

The tendency for feminist writers, especially during the 1970s and early 1980s, was to completely re-imagine Medusa as a subject in her own right, one that often reflected the anger animating so much of Second Wave feminism's struggle against gender inequality. As Susan Bowers writes, "contemporary women writers are turning to this matriarchal image for inspiration and empowerment. These artists demonstrate how the same image that has been used to oppress women can also help set women free" (Bowers 1990, p. 217). Here I give just a few prominent examples from this tradition. Mary Sarton's 1971 poem "The Muse as Medusa" closes with the quatrain, "I turn your face around! It is my face./That frozen rage is what I must explore—/Oh secret, self-enclosed, and ravaged place!/This is the gift I thank Medusa for" (Sarton 2003, lines 25–28). Earlier I quoted from Stanford's "Medusa," which sits firmly in this tradition of identifying Medusa with contemporary feminist rage. Towards the end of the poem, the speaker reflects on how she would like to move beyond the "fury that I did not choose" (Stafford 2003, line 31) but is unable to find peace because of memories—perhaps PTSD inspired flashbacks—of the rape by Poseidon (Stafford 2003, lines 33–37). As a final example, Emily Erwin Culpepper explicitly links the Medusa story with feminist resistance to patriarchal violence and oppression, making it a central theme in her 1986 essay with poetry "Ancient Gorgons: A Face For Contemporary Women's Rage." Culpepper recounts an attempt by a stranger to break into her apartment, and her feeling of power after fighting him off, which she aligns with the feminist Medusa (Culpepper 2003, pp. 242–45). She puts this experience into context: "The Amazon Gorgon face is female fury personified. This Gorgon/Medusa image has been rapidly adopted by large numbers of feminists who recognize her as one face of our own rage" (Culpepper 2003, p. 239). For Sarton, Stanford, and Culpepper, identification with Medusa becomes a liberatory gesture reflecting empowerment and the ability to turn a petrifying gaze upon patriarchy.

The impulse to reclaim a mythic figure such as Medusa is not necessarily an obvious one. As Vanda Zajko and Miriam Leonard put it, "many feminists have chosen to revivify ancient narratives to arm contemporary struggles. There is a tendency to overlook the strangeness of this choice. The myths are after all not only the products of an androcentric society, they can also be seen to justify its most basic patriarchal assumptions" (Zajko and Leonard 2008, p. 2). In other words, it is a striking irony that feminists use for liberatory ends a mythology steeped in anti-feminist ideology. However, this is part of a larger trend within canonical counter-discourse that Jeremy Rosen calls minor character elaboration, which he defines as, "constructing narratives around the perspectives of socially marginal figures in canonical works, often seeking to critique the ideologies underlying the manner in which these works represent minor characters—or their failure to represent socially marginal figures at all" (Rosen 2013, p. 139). In this genre—exemplified by works such as Jean Rhys' *Wide Sargasso Sea*, Tom Stoppard's *Rosencrantz and Guildenstern are Dead*, or J.M. Coetzee's *Foe*, for instance—authors identify characters who are silenced, sidelined, or otherwise peripheral because of their gender, ethnicity/race, social class, etc., and re-imagine their stories, often writing from the perspectives of those minor characters. This is essentially the project Goodman describes in the Introduction to *Mythic Women/Real Women*, which collects several performance pieces about women's experience, including mythological women. She says, "In the 1970s the focus was often on reading against the grain by looking for ways of re-examining male-created fictions so that the female figures trapped in these stories might come into sharper focus" (Goodman 2000, p. xii). This notion of freeing the voices or perspectives of trapped minor characters, with the accompanying notion that this helps liberate historically oppressed groups, is central to many feminist attempts to recover the Medusa story. However, as Rosen points out, there are substantial limits to the actual efficacy of minor character elaboration: "In lauding such texts as subversive or liberatory, critics transfigure their more modest political work of countering a previous representation with one that is more salutary. Lost is attention

to representation—to the fact that the depiction of a voice has been constructed by a contemporary author manifestly in possession of the cultural capital of the traditional canon who is writing on behalf of the formerly minor character" (Rosen 2013, p. 141). In other words, feminist poets or writers who assume the persona of Medusa do not actually give the mythic Gorgon a voice; Medusa is not empowered to speak, there is only the illusion of that empowerment created by a modern author imagining her perspective.

McCallum and Smartt's Medusa poems avoid the trap Rosen identifies with minor character elaboration by writing Medusa to help understand the speakers of their poems, rather than by identifying their speakers *as* Medusa. On perhaps the simplest level, this is evidenced by Smartt's continual third person references to Medusa, making clear that Medusa is never the speaker of the poem. This is in stark contrast to a poem such as Stanford's, where Medusa is presented in first person, with the speaker embodying Medusa as the I. In a larger sense, though, Smartt references Medusa to explore contemporary issues of racial representation, colonialism, and the exploitation of women. When Smartt addresses the painful self-erasure required for Black women to conform to white-created beauty standards in "medusa? medusa black!", this is clearly a contemporary concern. Smartt does not here speak "on behalf of" Medusa, does not attempt to give the mythical Gorgon a voice. Instead, Smartt's poet persona speaks for herself, using Medusa's reviled snaky hair as a metaphor for the shaming of Black women's hair in Europe and the US. The line "Banish the snake-woman" follows immediately after the string of images of straightening, burning, bleaching, and frying hair, linking Medusa's cursed locks to the way that white dominated societies conceptualize Black women's hair (Smartt 2001g, line 37). Similarly, the poem "let me land . . . " aligns Medusa with immigration from the Caribbean to Britain, a journey Smartt's own parents underwent. Being the child of Afro-Caribbean immigrants in London has profoundly shaped Smartt's experiences, as is reflected in her poetry.

With "Madwoman as Rasta Medusa," the line separating the speaker from the Gorgon is thinner, but still present. While McCallum's poem is in first person, it is from the perspective of Madwoman, a character who takes many forms and speaks with many voices throughout the poetry collection that bears her name. Because Madwoman shifts her identity throughout the collection, she may actually speak from the perspective of Medusa here—but if that is true, then it is equally true that Medusa speaks from the perspective of Madwoman. The speaker of this poem is clearly Jamaican, clearly a mystic, clearly Rasta. She uses Rasta apocalyptic rhetoric. This is, therefore, not the Medusa we know from Greek mythology. In other words, like Smartt, McCallum has not attempted to speak for the Medusa silenced in Hesiod, Apollodorus, Ovid, and other early writers; instead, McCallum's speaker blends her own identity with that of Medusa as a way of exploring shared experiences of oppression and exploitation.

## 6. Medusa Poems as Post-Colonial Canonical Counter-Discourse

These experiences of oppression and exploitation are, as we have already begun to see, deeply tied to race and to the post-colonial concerns of Caribbean authors. British colonial education systems imposed Greco-Roman classics and British literature, culture, and language on the colonized West Indies—as well as African and, to a lesser extent, South Asian colonies. Therefore, when Caribbean post-colonial authors such as McCallum and Smartt take up classical mythology or literature, they do so in ways that contest European cultural "ownership" of the classics. They engage, in other words, in canonical counter-discourse by writing back to the imperial center in such a way as to undermine imperial hierarchies.[11] Greenwood locates Caribbean uses of the classics in "a loaded cultural encounter between colonizer and colonized, [which] accords with the motif in Caribbean literature where the struggle for political and cultural autonomy is contested through the classics" (Greenwood 2005, p. 67). This struggle often involves re-imagining the position of classics as such within hierarchies of cultural authority. As Lorna Hardwick puts it, "analysis of rewriting of classical texts in colonised and colonising societies shows

that there is a pattern of features that suggest a distinctive role for classical material in provoking awareness and transformation of cultural identities" (Hardwick 2005, p. 107). In other words, post-colonial adaptations of the classics that write back to the center of imperial power/culture often do so to contest that centrality.

In their foundational work of post-colonial theory, *The Empire Writes Back*, Bill Ashcroft, Gareth Griffiths, and Helen Tiffin argue that language and cultural capital are central issues of contention in post-colonial literature and politics (Ashcroft et al. 1989, p. 7). Post-colonial writers such as McCallum and Smartt claim cultural ownership of Greco-Roman mythical figures such as Medusa, redeploying the Gorgon—as we have already seen—to make visible their own experiences as Caribbean women. However, even the mode of expression used by these poets represents a challenge to colonial/Eurocentric power structures. Language here literally functions as post-colonial resistance. According to Ashcroft, Griffiths, and Tiffin, "One of the main features of imperial oppression is control over language. The imperial education system installs a 'standard' version of the metropolitan language as the norm, and marginalizes all 'variants' as impurities" (Ashcroft et al. 1989, p. 7). This was what Ngũgĩ wa Thiong'o calls the "cultural bomb" of the British colonial education system in Africa (Ngũgĩ 2006, p. 3)—a system in which "The language of an African child's formal education was foreign . . . This resulted in the dissociation of the sensibility of that child from his natural and social environment, what we might call colonial alienation" (Ngũgĩ 2006, p. 17). British education systems in the colonial West Indies functioned similarly to those in Africa, with British and classical works being presented as *the* benchmark for literature and culture. In other words, the English language was used as a tool in African and Caribbean colonies to destroy locally rooted identities, and, specifically in the case of the Caribbean, to separate enslaved people from their native cultures, languages, and identities.

However, the politics of language are not merely oppressive, but also facilitate resistance and the formation of new identities in the face of Anglophone imperialism.[12] Both McCallum's "Madwoman as Rasta Medusa" and Smartt's poems reflect Caribbean versions of English—Jamaican Patois and Bajan Creole, respectively. These are languages developed amongst the common people, often the enslaved or the downtrodden, in the Caribbean, what Kamau Brathewaite calls *nation language*. Brathewaite's poetic description of nation language is "the submerged area of that dialect that is much more closely allied to the African aspect of experience in the Caribbean. It may be in English, but often it is in an English which is like a howl, or a shout, or a machine-gun, or the wind, or a wave. It is also like the blues" (Brathwaite 1993, p. 266). Historically, this was a language of resistance, challenging the primacy of standard English by constructing new West Indian folk cultures.[13] Cashman Kerr Prince calls this hybrid creation of new languages *post-colonial philology*, which he describes when "Language is created anew; this is not the search for linguistic origins or historical connections between languages. Rather, this is the process of creating new language and new connections between languages" (Prince 2010, p. 184). As with Ngũgĩ's turn to writing exclusively in Gĩkũyũ—his native Kenyan language—post-colonial philology is a form of resistance against imperial languages. Prince writes, "Breaking the customary order of language, reversing the accepted significations, marks a Creole writer's liberation from the hegemony of one language or culture, from the tyranny of a masterful imperial culture" (Prince 2010, p. 186). In other words, by rejecting standard English, with its colonialist history, and instead embracing a hybridized new language, post-colonial authors undercut the cultural authority of colonialism itself.

The use of Patois and Creole is evident throughout these poems. To take just the first line of McCallum's poem: "I-woman go turn all a Babylon to stone," this is clearly Patois, clearly Rasta-inspired (McCallum 2017, line 1). A standard English iteration might be: "I (a woman) am going to turn all of Babylon to stone." However, by using Patois, McCallum positions her speaker apart from the colonial power structures associated with standard English. The threat to Babylon—which, again, in Rasta theology can refer to a wide range of exploitative power structures—is actualized not only in the promise to petrify

it, but is enacted through a version of English developed out of Jamaican folk resistance to British colonialism.[14] In the same way, Smartt frequently deploys Bajan Creole rather than standard English, aligning the speaker of her poems with Barbadian folk culture. "medusa: cuts both ways," for instance, begins with the lines "Dread!/An Afrikanwoman/full of sheself/wid dem dutti-eye looks" (Smartt 2001e, lines 1–4). This opening overtly aligns the use of Creole language with African heritage by identifying the subject of the poem as an Afrikanwoman.

In addition to resisting a standard English imposed by British colonialism, these poems associate the cultural capital of the classics with the Patois and Creole languages. By discussing Greco-Roman mythology in Caribbean languages, these poems assert a cultural connection to ancient myth, assert that Caribbean culture is linked to antiquity in the same ways that colonial ideology would reserve for Europeans. This identification between the cultural capital of classics and Caribbean or Black diasporic culture is made most overt in Smartt's poem "medusa: cuts both ways," which explicitly identifies Medusa with a number of Black women, including Assata Shakur, Audre Lorde, Dorothy "Cherry" Groce, and Queen Nzinga Mbande (Smartt 2001e, lines 31–36).[15] The speaker then goes on to identify Medusa with a female African ancestral lineage and as a source of protection:

> Medusa is our mother's mothers
> myself all coiled into one
> Medusa is spirit
> Medusa in you is you in me
> is me in you
> Medusa is my shield
> Impregnable
> my aegis –
> no mythical aegeanpeople shield
> this is my armour
> with Shango double-headed axe
> Yemoja-Ocuti
> my battle dress armour
> of serious dread. (Smartt 2001e, lines 38–51)

These lines claim Medusa for African women, aligning her not only with individual African and diasporic women, as the previous lines had done, but symbolically identifying her as a collective ancestor. Smartt also almost adopts the position of Athena, claiming Medusa as the aegis, or divine shield of protection. However, she rejects the myth's Greek origin, renouncing "mythical aegeanpeople" in favor of locating Medusa within an African pantheon that includes the Yoruba thunder god Shango and water goddess Yemoja.[16]

There is a basis at least in Roman accounts of the Medusa myth for locating her in Africa, a tradition that Afro-Caribbean and other African diasporic writers who adopt the Medusa legend often (at least implicitly) capitalize on.[17] The Roman-era Greek writer Diodorus Siculus locates both the Amazons and the Gorgons in Africa, west of Egypt (Siculus 2003, p. 29). Lucan provides more details, writing that, "Medusa is said to have lived in the far west of Africa, at the point where the Ocean laps against the hot earth, in a wide, untilled, treeless region which she had turned entirely to stone merely by gazing around her" (Lucan 2003, p. 40). He also claims that there are poisonous snakes in Libya because they sprang up whenever Medusa's blood hit the ground as Perseus flew with her head over North Africa (Lucan 2003, p. 42). Given these accounts, it is reasonable to read post-colonial reclamations of Medusa as participating in the Afro-Classicist project, wherein classics scholars have documented the African and Middle Eastern/Asian roots of Greek culture. The most famous of the Afro-Classicists is probably Martin Bernal (1987), whose book *Black Athena* challenged traditional narratives about the origins of Greek civilization. In the context of artistic adaptation, Kevin Wetmore characterizes the "Black Athena" model as "a reappropriation of material that is already African in origin" (Wetmore 2003, p. 15).

While a poem such as "medusa: cuts both ways" may seem to reappropriate a myth that Diodorus Siculus and Lucan link to Africa, I would argue that McCallum and Smartt's poems both align more closely with Wetmore's "Black Dionysus" schema, which is another form of canonical counter-discourse. Wetmore explains the "Black Dionysus" model thus: "the Post-Afrocentric formulation … that is counter-hegemonic, self-aware, refuses to enforce dominant notions of ethnicity and culture, and uses ancient Greek material to inscribe a new discourse that empowers and critiques all cultures, even as it identifies the colonizer's power and the colonized's powerlessness" (Wetmore 2003, p. 44). In other words, the "Black Dionysus" style refuses essentialism, rather than attempting to re-inscribe a new Afro-centric, but still essentialist, model of classical culture. Instead, "Black Dionysus" writers refuse the notion that European cultures "own" Greco-Roman antiquity, laying claim to that heritage not because it is originally African, but because it is part of a shared global heritage. We see this in the hybridity of Patois and Creole language poems about classical mythology—McCallum and Smartt stake a claim to their (non-exclusive) ownership of the mythic canon, a claim embodied by their use of canonical counter-discourse.

## 7. Conclusions: The Stakes for Intersectionality in Canonical Counter-Discourse

Because modes of oppression, disenfranchising, and silencing frequently work along multiple planes, we must be aware of the multiple systems of oppression many subjects find themselves within. We cannot understand these experiences as unifacial. As female poets of color from the Caribbean, McCallum and Smartt face multiple axes of historical forms of oppression—gendered, racial, geographic, and as immigrants or children of immigrants. As Tiffin argues, canonical "counter-discursive strategies involve a mapping of the dominant discourse, a reading and exposing of its underlying assumptions, and the dis/mantling [*sic*] of these assumptions from the cross-cultural standpoint of the imperially subjectified 'local'" (Tiffin 1987, p. 23). While Tiffin is specifically interested in the post-colonial, subjects existing within dominant discourse are rarely one-dimensional, rarely just the colonized/post-colonial subject. Rather, subjects are also gendered, socio-economically classed, (dis)abled, aged, etc. Therefore, in order to effectively pursue the project of dismantling matrices of power and domination, canonical counter-discourse must be understood to be intersectional, resisting multiple axes of oppression within and between societies.

McCallum and Smartt model this intersectional canonical counter-discourse through the complexity of their Medusa poems. As we have seen, by utilizing Greco-Roman mythology, these writers draw on the cultural capital of classical antiquity to help authorize their own expression, and these expressions challenge both patriarchy and imperialist/racist metropolitanism. At the same time, however, the poems highlight the dynamics of power by presenting complex portraits that are or can be read as critical of Caribbean people as well. The "yu" to whom "Madwoman as Rasta Medusa" addresses itself could be patriarchal forces within Jamaica—in other poems Madwoman often encounters gendered oppression from the men in her life. Furthermore, Smartt's critiques of Black women painfully attempting to conform to white-centric beauty standards expose a degree of internalized racism within Black communities when natural African hair styles are rejected as ugly or unfashionable. In these poems, there is a clear linkage between the speakers' experiences as women and their experience as racialized/colonized subjects. To reduce the canonical counter-discourse of these Medusa poems to merely resisting one or the other form of oppression would be to miss a fundamental dimension of these authors' liberatory projects.

**Funding:** This research received no external funding.

**Institutional Review Board Statement:** Not applicable.

**Informed Consent Statement:** Not applicable.

**Data Availability Statement:** Not applicable.

**Conflicts of Interest:** This author declares no conflict of interest.

## Notes

[1]    Smartt originally presented *Medusa* as a one woman show, but I am examining only the printed poems from *Connecting Medium*—rather than the performance—for the sake of accessibility and synchronicity with McCallum's work.

[2]    Although *The Library* was written after *The Metamorphoses*, Apollodorus' version of Medusa is closer to Hesiod's than to Ovid's.

[3]    Roman equivalent to the Greek Athena. Neptune, named below, is the Roman version of Poseidon. When discussing Ovid—or modern poems using the same names—I use the deities' Latin names, otherwise I use the Greek names.

[4]    Feminist writers, especially in the twentieth and twenty-first centuries, draw heavily on this version, with the rape and the question of whether Medusa's transformation is a punishment or an empowerment as central thematic concerns. Below, this essay discusses some of these feminist thinkers and writers, including Hélène Cixous, Emily Culpepper, Mary Sarton, Doris Silverman, and Ann Stafford.

[5]    While this is a fairly widely accepted claim in reception studies, there is a school of thought which tries to pin down a core "real" version of the myth and rejects subsequent developments as aberrations. For instance, Leeming argues that only the Greco-Roman conception of Medusa can be seen as the "truth" of the myth. Addressing feminist critiques of the story, he writes, "our views do not change the reality of the myth. We are not the Greeks, and it is an ancient Greek myth" (Leeming 2013, p. 105).

[6]    Interestingly, in the 2021 Amazon Prime commercial "Medusa Makes Friends" (Amazon 2021) the titular character is explicitly presented as turning people to stone against her will—apart from the sleazy guy in the bar whom Medusa petrifies by lowering her sunglasses. The commercial also departs from classical descriptions of Medusa by showing her with a snake for her lower body. This was a characteristic not of Medusa but of Echidna, another Greek mythical monster.

[7]    Or sometimes just to men, a claim made by Doris K. Silverman (Silverman 2016, pp. 121–22). However, I do not know of any classical source that suggests Medusa only turned men to stone, sparing women. Silverman's reading is a psychoanalytical one. Following in the tradition of Freud and many other psychoanalyst critics who have engaged with the Medusa myth, Silverman finds in the myth what she wants to see, even if that requires repressing or re-imagining elements of the classical story.

[8]    The eight poems are "ten paces" (Smartt 2001i), "medusa? medusa black!" (Smartt 2001g), "medusa: cuts both ways" (Smartt 2001e), "dream bed" (Smartt 2001a), "medusa dream" (Smartt 2001f), "medusaspeak" (Smartt 2001h), "let her monsters write" (Smartt 2001c), and "let me land . . . " (Smartt 2001d).

[9]    I discuss "medusa: cuts both ways" in the subsequent section entitled "Medusa Poems as Post-Colonial Canonical Counter-Discourse."

[10]    Passing references to hair include lines 4 and 9 in "dream bed" (Smartt 2001a, lines 4 and 9), a mention at the opening of the last stanza in "medusa dream" (Smartt 2001f, line 63), line 8 of "let her monsters write" (Smartt 2001c, line 8), and the phrase "watershaking locks" in "let me land . . . " (Smartt 2001d, line 10).

[11]    In "Classics and the Atlantic Triangle," Greenwood (2004) argues that the triangle trade between Europe, Africa, and the Caribbean continues to provide a basic outline for interaction, but the modern flows are of cultural influence, rather than goods and people.

[12]    Similar issues exist within the Francophone Caribbean, where authors also have long traditions of both thinking critically about linuistic politics and adapting classical European texts. Authors such as Patrick Chamoiseau, Aimé Césaire, and Franz Fanon have written about colonial politics and language.

[13]    Ngũgĩ notes that the same kind of hybrid languages emerged in African colonies where English had been imposed: "when the peasantry and the working class were compelled by necessity or history to adopt the language of the master, they Africanised it without any of the respect for its ancestry shown by Senghor or Achebe, so totally as to have created new African languages . . . All these languages were kept alive in the daily speech, in the ceremonies, in political struggles, above all in the rich store of orature—proverbs, stories, poems, and riddles" (Ngũgĩ 2006, p. 23).

[14]    Many of McCallum's poems are written in standard English, but, as with "Madwoman as Rasta Medusa," she often writes in Patois. Similarly, Smartt writes in both standard English and Creole, depending upon the individual poem's purpose.

[15]    Assata Shakur was a member of the Black Panthers and then the Black Liberation Army, which advocated armed resistance against the oppression of African Americans. She is famous for having eluded the FBI since her 1979 escape from prison. Audre Lorde was a poet, artist, and scholar whose work has been influencial in several disciplines, including critical race studies, feminism, and queer theory. The 1985 Brixton Riot in London was sparked by the shooting of Cherry Groce, an innocent woman, by the Metropolitan Police. The office who shot her was subsequently acquitted of all charges and the Metropolitan Police never accepted liability. The Brixton Riot—like the Rodney King Riot several years later in Los Angeles and the #BlackLivesMatter movement in the 2000s—was a reaction to ongoing police violence against Black people. Nzinga Mbande was a seventeenth century ruler of two kingdoms in what is now Angola. A brilliant military leader, diplomat, and monarch, she resisted Portuguese attempts to colonize the region and is now remembered as a hero in Angola and many parts of the African diaspora.

<sup>16</sup> In Yoruba mythology, Yemoja is associated principally with the Ogun river. However, in many Western Hemisphere cultures—especially in Cuba and Brazil—Yemoja is linked to oceans.

<sup>17</sup> African American poets also frequently write about Medusa, including Countee Cullen's 1935 poem "Medusa" (Cullen 2003) and Collen J. McElroy's "A Navy Blue Afro," published in 1976 (McElroy 2003). McElroy's poem is thematically similar to Smartt's "medusa? medusa black!" in that both symbolically identify Black women's hair with Medusa's snakes, and the straightening, bleaching, etc., of it as a betrayal of African heritage.

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
