# Peer review of "Petrifyin’: Canonical Counter-Discourse in Two Caribbean Women’s Medusa Poems"

_humanities, doi:10.3390/h11010024_

Round 1

Reviewer 1 Report

This is an excellent essay and I believe it should be published with only minor revisions. I don't feel very equipped to judge the originality of the argument, but a cursory search (on google scholar) indicates that while there has been scholarship on the figure of the Medusa in Caribbean work and scholarship on these poets, no one person seems to have yet put these two women in conversation specifically for their use of Medusa imagery.  Below, I have included a short list of bullet-points where I would suggest some edits, but none of these are mandatory. 

-The opening line of the essay might be more striking: it wasn't a good indicator of how powerful the essay to come was. 

--Mention earlier the Jamaican/ Bajan heritage of the authors? They are just described as Caribbean until their individual sections.

--Could we go to the original Greek in line 145? What word is there for "all" in the translation?

--Should some more of the "feminist writers" who invoke the trope of the medusa be named more explicitly, if only in a footnote? Cixous springs to mind. Perhaps more specificity in note 4. 

--insist a bit more on the Medusa imagery "Banish the snake woman" in the poem presented on p. 7. The emphasis on beauty rituals in the analysis overwhelms the other very good point about agency and reclamation

--The language issues mentioned are also visible in the formerly French colonies and francophone Caribbean authors (like Patrick Chamoiseau) describe a similar battle with French Creole languages. Should this be given a note? (p. 10)

--Should the women claimed as ancestors in line 510 be given more of an introduction (in a note?) 

--Is Yemoja a river goddess? I associate her with the sea. or is this specific to Yemoja-Ocuti? (line 531) 

Author Response

See attached cover letter

Reviewer 2 Report

The article is a compelling reading of two Caribbean women poets, using Tiffin's theory of canonical counter-discourse to articulate their navigation of the Medusa myth as a vehicle for liberatory expression. The connections between the two poets are compelling and well-supported with quotations from the poems. References to key post-colonial theorists and literary frameworks advance the argument, and the historical references add a layer of depth to the analysis that is not present in all literary treatments of what Alicia Ostriker named "revisionist mythmaking." That is perhaps the one literary antecedent to the author's work that is most absent, but Ostriker's vein of analysis did not address the post-colonial and racialized aspects of the mythic treatments McCallum and Smartt create - so this is not necessarily a flaw in this article. But worth a comment to the author to look at Ostriker's work, given the other poets from the 1980s referenced in the piece, such as Stanford and Sarton.

Author Response

See attached cover letter
